Comparative analysis of shotgun metagenomics and 16S rDNA sequencing of gut microbiota in migratory seagulls

Liao Feng 1 2
Xia Yilan 3
Gu Wenpeng 4
Fu Xiaoqing 4
Yuan Bing 1 2 Bingyuan0109@163.com
1 Department of Respiratory Medicine, The First People’s Hospital of Yunnan Province , Kunming, Yunnan , China
2 The Affiliated Hospital of Kunming University of Science and Technology , Kunming, Yunnan , China
3 Department of Infectious Diseases and Hepatology, The First People’s Hospital of Yunnan Province , Kunming, Yunnan , China
4 Department of Acute Infectious Diseases Control and Prevention, Yunnan Provincial Centre for Disease Control and Prevention , Kunming, Yunnan , China
Gillespie Joseph
Electronic publication date: 2023 Nov 3
Publication date: 2023
Volume: 11
Electronic Location ID: e16394
Received 2023 May 15; Accepted 2023 Oct 11
Copyright: © 2023 Liao et al.
Copyright year: 2023
Copyright holder: Liao et al.
License: This is an open access article distributed under the terms of the Creative Commons Attribution License, which permits unrestricted use, distribution, reproduction and adaptation in any medium and for any purpose provided that it is properly attributed. For attribution, the original author(s), title, publication source (PeerJ) and either DOI or URL of the article must be cited.
License URL: https://creativecommons.org/licenses/by/4.0/

Keywords: Metagenome, 16S rDNA sequencing, Migratory seagulls, Gut microbiota, Comparative analysis

Funding: Joint Project of Applied Basic Research of Kunming Medical University 202001AY070001-291 Young and Middle Aged Academic and Technical Leaders Reserve Talent Plan 202105AC160033 Yunnan Provincial High Level Talent Training Plan YNWR-MY-2020-021 Yunnan Fundamental Research Project 202301AT070160 National Natural Science Foundation of China 82360398 This work was supported by the Joint Project of Applied Basic Research of Kunming Medical University (202001AY070001-291); the Young and Middle Aged Academic and Technical Leaders Reserve Talent Plan (202105AC160033); the Yunnan Provincial High Level Talent Training Plan (YNWR-MY-2020-021); the Yunnan Fundamental Research Project (202301AT070160) and the National Natural Science Foundation of China (82360398). The funders had no role in study design, data collection and analysis, decision to publish, or preparation of the manuscript.

==============================
Background

Shotgun metagenomic and 16S rDNA sequencing are commonly used methods to identify the taxonomic composition of microbial communities. Previously, we analysed the gut microbiota and intestinal pathogenic bacteria configuration of migratory seagulls by using 16S rDNA sequencing and culture methods.

Methods

To continue in-depth research on the gut microbiome and reveal the applicability of the two methods, we compared the metagenome and 16S rDNA amplicon results to further demonstrate the features of this animal.

Results

The number of bacterial species detected by metagenomics gradually increased from the phylum to species level, consistent with 16S rDNA sequencing. Several taxa were commonly shared by both sequencing methods. However, Escherichia, Shigella, Erwinia, Klebsiella, Salmonella, Escherichia albertii, Shigella sonnei, Salmonella enterica, and Shigella flexneri were unique taxa for the metagenome compared with Escherichia-Shigella, Hafnia-Obesumbacterium, Catellicoccus marimammalium, Lactococcus garvieae, and Streptococcus gallolyticus for 16S rDNA sequencing. The largest differences in relative abundance between the two methods were identified at the species level, which identified many pathogenic bacteria to humans using metagenomic sequencing. Pearson correlation analysis indicated that the correlation coefficient for the two methods gradually decreased with the refinement of the taxonomic levels. The high consistency of the correlation coefficient was identified at the genus level for the beta diversity of the two methods.

Conclusions

In general, relatively consistent patterns and reliability could be identified by both sequencing methods, but the results varied following the refinement of taxonomic levels. Metagenomic sequencing was more suitable for the discovery and detection of pathogenic bacteria of gut microbiota in seagulls. Although there were large differences in the numbers and abundance of bacterial species of the two methods in terms of taxonomic levels, the patterns and reliability results of the samples were consistent.

Introduction

The microbiomes of different host organisms are currently described by using culture-independent sequencing methods (Harvey & Holmes, 2022). Shotgun metagenomic and 16S rDNA sequencing methods are commonly used to identify the taxonomic composition of microbial communities in several studies (de Melo et al., 2020; Mackelprang et al., 2011; Monaco & Kwon, 2017; Qin et al., 2010). Shotgun metagenomic sequencing of total DNA in a sample increases the resolution of species-level taxonomic assignments and provides direct gene functional annotations. It can also be used to analyse the nonbacterial parts of microbiomes, such as viruses and fungi. The high-resolution assemblies provide in-depth insight into microbial diversity (Warnecke et al., 2007). 16S rDNA sequencing is a cost-effective and straightforward method to investigate the microbiota of different organisms. However, it has limited utility because of the length of the amplicon of the target gene. This method can only provide the bacterial members of microbial communities and brings about some sequence artifacts or bias in taxonomic profiles (Franzosa et al., 2015). Rausch et al. (2019) compared amplicon and metagenomic sequencing methods in animal metaorganisms. They demonstrated that many aspects of bacterial community characterization were consistent across the two sequencing methods. Their results facilitated the selection of appropriate methods across a wide range of host taxa (Rausch et al., 2019). Ranjan et al. (2016) compared the whole genomic shotgun sequencing with 16S rDNA method for human fecal microbiome. They found that shotgun metagenomics had multiple advantages compared with the 16S amplicon method, such as enhanced detection of bacterial species, increased diversity and prediction of genes (Ranjan et al., 2016). Therefore, both sequencing methods have their own advantages and disadvantages and are widely used in current biological studies.

The black-headed seagull (Larus ridibundus, hereafter seagull) is a migratory wild bird worldwide at present. The public is in close contact with migratory seagulls when they migrate from Siberia to southern China to overwinter at parks or rivers of the cities (Frixione et al., 2022; Zeballos-Gross et al., 2021). Previously, we analysed the features of gut microbial communities and pathogens of this animal by 16S rDNA amplicon sequencing and culture methods (Liao et al., 2019). The results indicated that many pathogenic bacteria, such as enteropathogenic Escherichia coli and Salmonella spp. were isolated, although little cross infection was found between humans and seagulls (Liao et al., 2019). However, amplicon sequencing and culture method only reflect limited findings regarding the microbiota of animals. Currently, advanced methods have been used to perform in-depth analyses on the connection between bird activities and disease dissemination; in particular, metagenomics provides us with a broader characterization of microbe diversity (Ramirez-Martinez et al., 2018). In addition, assessing the taxonomic diversity of microbes is a key point to understanding the interaction between microbes and hosts (Wille et al., 2021). To date, there is no direct comparison of the utility of the two methods mentioned above in the gut microbes of migratory seagulls. In this study, we compared the effects of profiling the seagull gut microbiome with metagenomic vs 16S rDNA sequencing in detail to further explore the tradeoffs of each method.

Materials and Methods

Sample collection

A total of 160 seagull fecal samples were collected from November 2021 to January 2022. Samples were collected at two locations (HuanXiQiao, HXQ and HaiGeng, HG), and 80 samples were collected at each sampling location. HXQ (25.04 N, 102.69 E) is a central park in an urbanized area. HG (24.96 N, 102.65 E) is the surrounding area of Dianchi Lake in Kunming city, which is a more rural area. The distance between the two sampling locations was approximately 11 km. Fresh feces of seagulls were collected and stored in sterile containers. Twenty samples of each group were mixed into one for subsequent experimental analysis, and the mixed samples were collected approximately 5–10 m from each other for each sampling location.

Metagenomics

Genomic DNA was extracted using fecal sample total genomic DNA extraction kits (Tiangen, Beijing, China) according to the manufacturer’s instructions. The DNA quality was detected using Qubit 2.0 (Thermo Fisher Scientific, Waltham, MA, USA) and Nanodrop accordingly. Quantified genomic DNA was fragmented by sonication to a size of 350 bp and then end-repaired, A-tailed, and adaptor ligated using the NEB Next DNA Library Prep Kit for Illumina (NEB, Ipswich, MA, USA) according to the instructions (Pessoa et al., 2013; Pham, Palden & DeLong, 2007; Solonenko & Sullivan, 2013). DNA fragments with lengths of 300–400 bp were enriched by PCR (12 cycles) and purified using an AMPure XP system (Beckman Coulter, Brea, CA, USA). The libraries were analysed for size distribution by a 2100 Bioanalyzer (Agilent, Santa Clara, CA, USA) and quantified using real-time PCR. Finally, genome sequencing was performed on an Illumina NovaSeq 6000 by a paired-end 150 (PE150) reagent kit.

16S rDNA sequencing

Genomic DNA was extracted as mentioned above, and the samples were identical to metagenomic sequencing. The V3-V4 variable region of the 16S rDNA gene was amplified using a previously described method (Gu et al., 2021). In general, PCR was performed using the KAPA HiFi Hot Start kit (Kapa Biosystems, Wilmington, MA, USA). The amplification procedure has been previously described (Liao et al., 2019). The AMPure XP beads (Beckman Coulter, Brea, CA, United States) was used to purify and quantify PCR products. The Illumina Nextera barcodes was added by secondary PCR procedure, and then the products were purified again to clear the nontarget fragments. The normalized and pooled amplicons were used for sequencing by using an Illumina NovaSeq 6000 system (Illumina, San Diego, CA, USA).

Statistical analyses

For metagenomics, raw data containing adapters or low-quality reads were filtered according to the following rules using FASTP (Chen et al., 2018) (version 0.18.0): (1) reads containing more than 10% unknown nucleotides (N) were removed; (2) reads containing less than 50% bases with quality (Q-value) >20 were removed; and (3) reads aligned to the barcode adapter were removed. After filtering, the resulting clean reads were used for genome assembly. Clean reads of each sample were assembled individually using MEGAHIT (version 1.1.2) stepping over a k-mer range of 21 to 141 to generate sample (or group)-derived assembly (Li et al., 2015). Genes were predicted based on the final assembly contigs (>500 bp) using MetaGeneMark (version 3.38) (Zhu, Lomsadze & Borodovsky, 2010). The reads were aligned to predict genes using Bowtie (version 2.2.5) to count read numbers (Langmead & Salzberg, 2012). The final gene catalogue was obtained from nonredundant genes with gene read counts >2. The plots were graphed using the R ggplot2 package (Gustavsson et al., 2022). The unigenes were annotated using DIAMOND (version 0.9.24) (Buchfink, Xie & Huson, 2015) by alignment with the deposited unigenes in different databases, including Nr, KEGG, eggNOG, CAZy, PHI, VFDB, and CARD. Since microorganisms such as fungi and eukaryotes would be detected in the metagenome, to ensure the comparability of the 16S with the metagenomics, we only retained the annotations as bacteria in the metagenome to participate in the analyses and recalculated the relative abundance of each taxon by treating the abundance of bacteria as a total (100%). Specifically, we removed results annotated as fungi and eukaryotes in each metagenomic sample. The remaining results annotated as bacteria were considered as a whole, and the relative abundance of each taxonomic level in each sample was recalculated.

Chao1, ACE, Shannon, and Simpson indices were calculated using Python language (version 0.5.6). Rarefaction defined at the 97% sequence similarity cut-off value was used to evaluate whether the sequencing was sufficient to cover all taxa. Alpha diversity comparison between groups was calculated by the Wilcoxon rank test. A Bray‒Curtis distance matrix based on gene/taxon/function abundance was generated separately by the R Vegan package (Huson et al., 2007). The multivariate statistical technique of PCoA was calculated and plotted using the R ggplot2 package. The Anosim test was calculated using the R project Vegan package. Heatmap graphs were plotted using the R heatmap package.

Bowtie2 and bedtools (version 2.29.0) were used to cluster all contigs longer than 1.5 kb from all samples into bins with default parameters (Alneberg et al., 2014). Then, contigs of samples were clustered to recover metagenome-assembled genomes using MetaBAT2 with default parameters (Kang et al., 2019). Finally, CheckM (version 1.0.6) was applied to estimate the completeness and contamination for each bin (Parks et al., 2015). Finally, the bins with high completeness (>80%) and fewer contaminants (<5%) were retained as draft genomes. The contamination was estimated from the number of multicopy marker genes identified in each marker set.

For 16S rDNA sequencing, the raw data were trimmed and quality controlled according to the rules using FASTP (version 0.18.0) as mentioned above. The paired-end reads were merged, and barcodes were removed using PEAR 0.9.6, cutadapt 1.2.1, and Prinseq 0.20.4. QIIME 2.0, USEARCH 11.0, and R 3.2 were used for bioinformatics analysis (Gu et al., 2020). The chimeric tags were removed, and effective tags were used. Sequences were clustered into operational taxonomic units (OTUs) according to 97% sequence similarity against the Silva 132 database using the UPARSE pipeline (Edgar, 2013). OTUs were named using SILVA taxonomic nomenclature. LEfSe analysis was used to identify bacteria with statistically significant differences in abundance between the two methods (Chang, He & Dang, 2022). PCoA was used to visualize the similarities between the HXQ and HG groups. ANOSIM was used to compare the differences in microbial communities between the HXQ and HG groups. PICRUSt was used to predict the functional potential of the communities (Douglas, Beiko & Langille, 2018).

The numbers of common and unique bacterial OTU between the two methods were visualized in a Venn diagram by the R package (Chen & Boutros, 2011). The top 10 average abundances of taxa at different taxonomic levels were compared between the two methods and visualized by a stacked graph (Conway, Lex & Gehlenborg, 2017). The Pearson correlation coefficient of the two methods was calculated using the R psych package, and the correlation significance was calculated using Fizh-Z transformation (Oliveira et al., 2020). Scatter plots were used to show linear correlations between the two methods at different levels of taxa. Beta diversity of the two methods was compared with the Mantel test by the R vegan package (Segata et al., 2011). Procrustes analysis based on the spatial distribution of PCoA of samples was used to determine the degree of correlation between the two methods (Chernoff & Nielsen, 2009).

Negative control and contamination exclusion experiments have been reported in our previous study (Liao et al., 2023). All python scripts used in the study were submitted to dryad (DOI 10.5061/dryad.w9ghx3fvw).

Availability of data

All data generated or analysed during this study are included in this article. The sequence data have been deposited into the National Center for Biotechnology Information (NCBI), https://www.ncbi.nlm.nih.gov/ with BioProject accession number: PRJNA849401 for metagenome and PRJNA849758 for 16S rDNA sequencing.

Results

The average effect reads after quality control for the metagenome of total samples was 99.74% ± 0.05% and 90.09% ± 4.15% for the 16S rDNA sequencing method. The composition of the gut microbiome of the metagenome indicated that 99.72% was bacteria, followed by viruses (0.2%), fungi (0.07%), archaea (0.001%) and eukaryotes (0.0002%). Annotations as bacteria in the metagenome were recalculated for the relative abundance of each taxon for comparison with 16S rDNA sequencing. The total abundance of all bacteria was compared at different taxonomic levels for both sequencing methods. Statistics of common and unique OTUs at the phylum level indicated that 13 phylum taxa were shared both with metagenome and 16S rDNA sequencing, 11 and seven unique taxa existed for metagenome and 16S rDNA sequencing, respectively (Fig. 1A). At the class level, 15 taxa were shared with both methods, but 31 and 19 unique taxa for each of them (Fig. 1B). At the order level, 32 common taxa were shared by the two methods, with 61 and 33 unique taxa for each method (Fig. 1C). At the family level, there were 61 common taxa but 118 and 49 unique taxa, respectively (Fig. 1D). The largest differences were found at the species level; only 13 species taxa were shared with both methods, while 1,568 and 67 taxa were identified for metagenome and 16S rDNA sequencing, respectively, as shown in Fig. 1E. The cladogram showed that commonly shared taxa were Firmicutes, Proteobacteria, Actinobacteria, Bacteroidetes and Cyanobacteria at the phylum level of the two methods. Bacilli and Gammaproteobacteria at the class level; Enterobacterales, Lactobacillales and Pseudomonadales at the order level; Enterobacteriaceae, Enterococcaceae, Lactobacillaceae, Moraxellaceae, Pseudomonadaceae at the family level; and Aerococcus, Catellicoccus, Lactobacillus, Acinetobacter, Psychrobacter and Pseudomonas at the genus level were the shared taxa of metagenome and 16S rDNA sequencing (Fig. 1F and Table S1). Erwiniaceae, Hafniaceae, Escherichia, Shigella, Erwinia, Escherichia albertii, Escherichia coli, Shigella sonnei and Erwinia gerundensis were unique taxa for the metagenome compared with Escherichia-Shigella, Hafnia-Obesumbacterium, Lelliottia and Pantoea for 16S rDNA sequencing (Fig. 1F and Table S1).

Figure 1 Comparison of the metagenome and 16S rDNA sequencing results for the number of bacterial taxa recovered by each method from fecal samples of the black-headed seagulls at the following levels of classification.

Binning analysis results indicated that four high-quality and 87 low-quality bins were generated from all samples (Fig. S1A). The features of high-quality bins were shown in Fig. S1B. Bin_ALL.85 had the highest contig sequencing depths compared with Bin_ALL.10, which had the lowest. However, Bin_ALL.10 showed the highest GC content compared with the others. The annotation results of bins at the genus level demonstrated that Pseudomonas spp., Pantoea spp., Kocuria spp. and Lactobacillus spp. were the taxonomies corresponding to Bin_ALL.77, Bin_ALL.85, Bin_ALL.10 and Bin_ALL.46, respectively (Fig. S1C).

The top 10 relative abundances of different taxonomic levels of metagenome vs 16S rDNA sequencing were shown in Fig. 2. At the phylum level, Proteobacteria, Firmicutes, Actinobacteria and Bacteroidetes accounted for 98.53%, 1.42%, 0.048% and 0.008% of the metagenome for all the samples compared with 88.89%, 10.62%, 0.27% and 0.21% of 16S rDNA sequencing (Fig. 2A). At the class level, Gammaproteobacteria and Bacilli accounted for 98.53% and 1.40% of the metagenome compared with 88.92% and 10.46% of 16S rDNA sequencing (Fig. 2B). Enterobacterales (92.61%), Pseudomonadales (5.99%) and Lactobacillales (1.34%) were the most abundant taxa at the order level of the metagenome, while 74.41%, 14.56% and 10.53% of the relative abundance of the 16S rDNA sequencing results corresponded to each taxon (Fig. 2C). The most relative abundance of metagenome at family level were Enterobacteriaceae (90.08%), Moraxellaceae (5.90%) and Erwiniaceae (3.30%). However, Enterobacteriaceae (75.69%), Moraxellaceae (7.76%), Pseudomonadaceae (7.03%), Enterococcaceae (4.06%) and Lactobacillaceae (3.52%) were the most distributed taxa in the 16S rDNA amplicon results (Fig. 2D). Escherichia (80.45%), Psychrobacter (8.09%), Shigella (3.69%), Erwinia (2.75%), Klebsiella (1.10%), and Salmonella (0.74%) were the most relatively abundant genera in the metagenome, while Escherichia-Shigella (61.91%), Pseudomonas (7.11%), Pantoea (5.70%), Psychrobacter (5.33%), Lelliottia (5.12%), Catellicoccus (4.08%), Lactobacillus (3.57%) and Hafnia-Obesumbacterium (2.70%) were the most abundant genera in the 16S rDNA sequencing (Fig. 2E). Finally, the largest differences were identified at the species level of the two methods. In general, Escherichia coli (73.85%), Escherichia albertii (12.10%), Shigella sonnei (4.36%), Erwinia gerundensis (4.13%), Salmonella enterica (1.09%) and Klebsiella pneumoniae (0.94%) were the most abundant taxa of the metagenome at the species level, compared with Psychrobacter frigidicola (32.69%), Catellicoccus marimammalium (27.15%), Lactobacillus aviaries (14.36%) and Pseudomonas fragi (12.65%) according to 16S rDNA sequencing (Fig. 2F). The total abundance of taxa for all samples was shown in Table S2.

Figure 2 Comparison of the relative abundance of the metagenome and 16S rDNA sequencing results in this study.

Alpha diversity analysis showed statistical significance between the HG and HXQ groups of the metagenome for the Ace, Chao1, Shannon and Simpson indices (Figs. 3A–3D upwards). Similar results could be identified between the two groups of 16S rDNA sequencing for all the indices (Figs. 3A–3D downwards). Beta diversity revealed that high diversity was found in the HG group compared with the HXQ group by metagenomic and 16S rDNA sequencing methods (Figs. 4A and 4B), but no significant differences were found. The details of the diversity statistical results between the HG and HXQ groups were shown in Table 1.

Figure 3 Comparison of alpha diversity for metagenome and 16S rDNA amplicon results.

Figure 4 Comparison of beta diversity for metagenome and 16S rDNA amplicon results.

Table 1 Diversity statistical results between HG and HXQ group in this study.

	Index	P value	
Metagenome	16S rDNA sequencing	
α-diversity	Ace	0.0078	0.020	
	Chao1	0.0078	0.014	
	Shannon	0.0078	0.020	
	Simpson	0.0078	0.020	
β-diversity	Anosim	0.222	0.110	

Pearson correlation analysis of the two sequencing methods indicated that statistical significance could be found at the phylum/family/genus/species levels. However, the R value of consistency gradually decreased with the refinement of the taxonomic levels (Figs. 5A–5D). A high correlation coefficient (r = 0.8701) was found at the phylum level compared with a low correlation coefficient (r = 0.0515) at the species level. Beta diversity by the Mantel test revealed that there was no statistically significant correlation at the phylum and species levels, but significant correlations were identified at the family and genus levels (Figs. 5E–5H). Specifically, a high correlation coefficient (r = 0.936) was found at the genus level. Similar results could be found for the procrustes tests of the two sequencing methods (Figs. 5I–5L). Significant statistical correlations were recognized only at the family and genus levels, with the highest correlation coefficient (r = 0.981) at the genus level.

Figure 5 Correlation analysis and beta diversity comparison of metagenome and 16S rDNA sequencing results in this study.

Discussion

Bird diseases such as avian influenza, avian cholera and West Nile fever have occurred frequently since the 20th century, leading to the death of wild birds, poultry and humans (Craft, 2015). Birds have strong environmental adaptability and bring potential risks to the dissemination of diseases. Several studies have characterized the relationship between bird migration and disease transmission (Jarma et al., 2021; Phan et al., 2013; Price et al., 2016). In particular, our previous study revealed that Enterococcaceae, Enterobacteriaceae, and Mycoplasmataceae were the most major families of gut microbial communities of seagulls by using 16S rDNA sequencing. Enteropathogenic E. coli and Salmonella were the most isolated pathogenic bacteria of this animal, although little cross infection was found between humans and seagulls by PFGE method (Liao et al., 2019). Metagenomic and 16S rDNA sequencing are the most commonly used methods to identify the features of microbial communities. Several studies have demonstrated their usage and applicability in different study areas but have not referred to migratory seagulls of gut microbiomes. Therefore, we revealed the similarities and differences between the two methods in this study to further explore the characteristics of seagull gut microbial communities.

16S rDNA amplicon sequencing is widely used in the field of microorganisms. It has a lower cost and is the preferred omics method for microbial community analysis (Klindworth et al., 2013). However, the amplified fragments of this method are approximately 400 bp, which limits its application. In this study, we found that the bacterial OTUs composition and alpha/beta diversity of the gut microbes of seagulls were similar by using metagenomic and 16S rDNA sequencing. As the taxonomic hierarchy of bacterial OTUs moved from the phylum to the species level, the differences between the two methods gradually increased. Without considering relative abundance, there were 61 taxa detected by both methods at the family level, only 49 taxa unique to the 16S amplicon, and 118 taxa unique to the metagenome. Furthermore, this difference was even larger at the species level. It was considered that the identification range of bacterial OTUs was greatly limited due to primer amplification and short sequencing lengths, the more refined taxonomic levels, and the greater limitations of 16S rDNA sequencing (Eloe-Fadrosh et al., 2016).

The abundance of taxa also influenced the results of differences. At the phylum level, the correlation coefficient between the two sequencing methods was up to 0.8701. However, at the genus level, the correlation coefficient was only 0.3215. Although increasing the threshold for the abundance filter might improve the correlation, the overall difference still existed. In general, relatively consistent patterns and reliability could be identified by both methods, but the results varied following the refinement of taxonomic levels. Taking both abundance and diversity into consideration, even at the genus level, the two sequencing methods still had a highly significant correlation of 0.936 (Mantel test). This indicated that although there were large differences in the numbers and abundance of bacterial OTUs of the two methods in terms of taxonomic levels, the final results of the samples were consistent. Based on the comparative analysis, we considered the 16S rDNA sequencing method to be the “sparrow type”, with low cost, wide species coverage, and reliable abundance assessment. Metagenomics was recognized as the “peacock type”, with more powerful data analysis ability (Schloissnig et al., 2013). Similar findings have been reported in previous studies (Peterson et al., 2021; Ravi et al., 2018; Rubiola et al., 2022). They also demonstrated that many aspects of bacterial community characterization were consistent across methods (Rausch et al., 2019). Therefore, both metagenomic and 16S sequencing were effective methods to analyse the microbial community, and the method selection possibly depended on different research purposes.

However, it was worth noting that at the species level, the metagenomic sequencing approach was able to detect many pathogenic bacteria, such as E. albertii, S. sonnei, S. enterica, K. pneumoniae and S. flexneri. In contrast, 16S rDNA sequencing was not as effective in detecting pathogenic bacteria and hardly detected any bacteria that were pathogenic to humans. Choi et al. analyzed the Salmonella prevalence and pathogen-microbiota relationships in Barn Swallows by using 16S rDNA sequencing and culture methods. They found that 16S rDNA sequencing was better than culture method for detecting Salmonella, and highlighted the value of 16S rDNA gene sequencing for monitoring pathogens in wild birds (Choi ON et al., 2021). Therefore, from this point of view, the metagenomic sequencing method was more suitable for the discovery and detection of pathogenic bacteria, although in general, the results of the two sequencing methods were consistent.

Conclusions

In this study, we compared the metagenome and 16S rDNA amplicon results to further demonstrate the features of migratory seagulls of gut microbiomes. Relatively consistent patterns and reliability could be identified by both methods to demonstrate the characteristics of gut microbial communities of seagulls, but the results varied following the refinement of taxonomic levels. Metagenomic sequencing was more suitable for the discovery and detection of pathogenic bacteria of gut microbiota in seagulls. Although there were large differences in the numbers and abundances of the two sequencing methods in terms of taxonomic levels, the final results of the samples were consistent.

Supplemental Information

Supplemental Information 1 The original data of cladogram for common and unique taxa composition between the two methods.

Click here for additional data file.

Supplemental Information 2 The total taxa abundance of all samples in this study.

Click here for additional data file.

Supplemental Information 3 The binning results of the metagenome from all samples.

Click here for additional data file.

We are grateful to Guangzhou Gene Denovo Biotechnology Co., Ltd. for assisting in the bioinformatics analysis. We sent our manuscript to American Journal Experts for English language revision.

Additional Information and Declarations

Competing Interests

Author Contributions

Data Availability

The authors declare that they have no competing interests.

Feng Liao performed the experiments, prepared figures and/or tables, and approved the final draft.

Yilan Xia performed the experiments, authored or reviewed drafts of the article, and approved the final draft.

Wenpeng Gu analyzed the data, prepared figures and/or tables, and approved the final draft.

Xiaoqing Fu analyzed the data, authored or reviewed drafts of the article, and approved the final draft.

Bing Yuan conceived and designed the experiments, authored or reviewed drafts of the article, and approved the final draft.

The following information was supplied regarding data availability:

The datasets for this study are available at the National Center for Biotechnology Information (NCBI): PRJNA849401 for metagenome and PRJNA849758 for 16S rDNA sequencing.

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
