# Peer review of "Comparative analysis of shotgun metagenomics and 16S rDNA sequencing of gut microbiota in migratory seagulls"

_PeerJ, doi:10.7717/peerj.16394_

## Round 0.1 · original submission · Major Revisions

Dear Dr. Liao and colleagues:

Thanks for submitting your manuscript to PeerJ. I have now received three independent reviews of your work, and as you will see, one reviewer recommended rejection, while the others suggested major revisions. I am affording you the option of revising your manuscript according to all three reviews but understand that your resubmission may be sent to at least one new reviewer for a fresh assessment (unless the reviewer recommending rejection is willing to re-review).

In general, the reviewers wish to see a much-improved manuscript as far as presentation and clarity. There are a lot of suggestions that will help here, but you should treat the revision as a major restructuring and presentation. Importantly, the English and grammar need a major overhaul; for example, reviewer 1 pointed out many problems (but this was not exhaustive, and you should seek assistance with English and grammar and copyediting).

There appears to be literature that needs citing. The conclusions should be assessed considering all the appropriate studies in the field. Please address concerns about subjectivity (in particular, why certain analyses were performed over others). Also, ensure that the methods are clearly presented, justified, and able to be repeated in their entirety.

There are many minor problems pointed out by the reviewers, and you will need to address all of these and expect a thorough review of your revised manuscript by these same reviewers. I agree with the concerns of the reviewers, and thus feel that their suggestions should be adequately addressed before moving forward.

Therefore, I am recommending that you revise your manuscript, accordingly, taking into account all of the issues raised by the reviewers.

I look forward to seeing your revision, and thanks again for submitting your work to PeerJ.

Good luck with your revision,

-joe

Reviewer 1 ·

Basic reporting

Overall this manuscript comparing 16S and metagenomic sequencing will be useful, especially to those identifying the most appropriate sequencing method for their study. However, there are several major issues which need to be addressed throughout the manuscript.

One primary concern is that throughout this manuscript, bacterial "species" are frequently referred to. In some places (such as line 206) species is incorrectly used. In other places, the use of species may be misplaced. Few bacterial species are defined, compared to the estimated number of likely true species. I suggest using a different word or phrase in place of species. The authors clustered sequence variants into 97% OTUs. Perhaps OTU is a better word that species in many instances (such as line 209).

Experimental design

Methods – a description of if and how contaminants were removed from the datasets is needed. Similarly were cyanobacteria, archaea, eukaryotic DNA, etc removed? In the taxonomic barpots figure a substantial amount of “other” and “unclassified” is included.

It might be easier to follow the methods if all statistical analyses were reported together, instead of some under the metagenomics heading and some under the 16S heading. It may also help to write why specific methods were being used instead of simply writing the method (for example line 192).

Validity of the findings

Results – Why no reporting on bacteria Class?

I’m confused as to why there are 13 phyla shared between both sequencing methods but only 2 are discussed in any kind of detail.

A supplementary file of taxa abundance should be included

A table including statistical results should be included

Discussion
I would like to see some discussion on the differences between the sequencing methods. Were specific taxa only recovered from one sequencing method of particular note, such as pathogenic bacteria? Possibly making one sequencing method or the other more applicable to specific research questions.

Additional comments

Line 54 – What species?

Line 56 – I’m confused by this sentence. First are the authors referring to number of bacterial species? If so please specify. Second, the number of species recovered for metagenomic or 16S sequencing should remain consistent regardless of what taxonomic level is being examined.

Results paragraph – The use of species here will be contentious for many people familiar with microbiome research. Few bacterial species are actually defined and named. Perhaps the author means ASVs, defined species or genera?

Line 71 – Please clarify what final results were consistent.

Line 89 – I’m guessing the author means The different HOST organisms. If so please clarify.

Line 98 – I recommend replacing resolution with utility or a similar word.

Line 103 – Suggested edits:Black-headed seagull (Larus ridibundus, hereafter seagull) is a migratory wild bird that has become one of the most popular species in…

Line 105 – at parks not on parks

Line 109 – please identify previous study

Line 112 – Suggested grammatical edit: advanced methods have been used to perform in-depth analyses

Line 115 – Not sure keystone is the correct word here, also understand should be understanding

Line 119 – detail not details

Line 124 – If possible spell out what HXQ and HG mean the first time these acronyms are used.
Line 124 – at not according

Line 125 – Living environment of the human is ambiguous, this sentence could be rephrased as HXQ is a central park in an urbanized area. HG is the surrounding area of Dianchi Lake of Kunming city, which is a more rural area.

Line 128 – Remove Every and spell out 20 (Twenty samples of each group…)

Line 133 – Please include which specifically which DNA extraction kits were used.

Line 135 – Should qualified be quantified?

Line 133-142 – Were these methods the same for both 16S and metagenomic sequencing?
Line 153 – include and between Shannon and Simpson; replace index with indices.

Line 154 – Alpha index could be replaces with Alpha diversity

Line 155 – if gene/taxon/function abundance comparisons were conducted separately please note that.

In methods, the authors switch between using passive and active voice. Please make this consistent throughout.

Line 165 – How were contaminants identified?

Line 168 – “and the samples were paired with metagenomics”. Please clarify this.

Line 170 – previously described? Referenced? Primers. Please clarify

Line 177 – Please describe how raw data were trimmed and qhat quality control measures were used.

Line 180 – Please include justification for 97% OTU clustering

Line 182 – What groups were used in LEfSe analysis and PCoA and anosim?

Line 187 – In paragraph heading or within the paragraph, please specify which methods are being compared.

Line 188 – What statistics? Also, I recommend using visualized instead of shown.

Line 195 – different levels of what?

Line 208 – Unless I am misunderstanding the results, I do not think using only bacterial data from metagenome is appropriate.

The reporting of the results is confusing to follow. The authors say “common and unique species at phylum level”. Are they comparing total abundance/presence of all bacteria within a specific phylum or are they only comparing specific species within a specific phylum. This should be clarified throughout this paragraph.

Line 237 – What does distributed mean here? Same with line 246

Line 251 – index should be indices.

Line 254 – please clarify what both methods are.

In general, the methods are hard to follow. “Methods” or “both methods” is often ambiguous as to if the authors are referring to statistical methods or sequencing methods. Also, taxonomic level being analyzed and reported should be clarified throughout.

Line 268 – represented by could be changed to “such as” or “including”

Lin 272 – instead of showed use characterized or describe; instead of concerned used focused.

Line 282 – mostly used methods should be most commonly used methods

Line 288 – extremely high cost performance is unclear.

Line 290 – limits not limited

Line 302 – what is meant by consistency here?

Line 303 – Correlation probably isn’t the correct word here. Similarity?

Line 308 – How is correlation being calculated? 0.936 is quite different than 0.3215

Figure 1 – meta16S could be just 16S and metaGenome could be metagenome – in this figure and throughout

Figure 2 – what does Bin_id refer to? Here and through the manuscript.
Was contamination removed from datasets before running analyses?

Figure 3 – Visualization of taxonomic barplots would be easier if only the most common groups were reported. For example in 3A many phyla are included in the legend but not seen (as far as I could tell) within the barplots.

I recommend double checking the OTUs identified as unclassified or other as these might not be true bacteria (could possibly be host DNA) and removing those from the figures and from analyses.

Figure 4 – The legends for the metagenome row could be removed
I am surprised that the p value is so high for the 16S comparisons, the boxplots do seem to vary quite a bit.

Figure 5 and 6 legends need more information

Reviewer 2 ·

Basic reporting

This manuscript focuses on comparing the gut microbiota of seagulls using shotgun and 16S rRNA metagenomics. Analysis of seagulls gut microbiota can be an important bioindicator to carry out zoonoses surveillance as they are commonly found in urban areas where they scavenge food discarded by humans.

The English of the manuscript should be improved and further revised before it is published. Some examples where the English could be improved include, among others, the title, I could suggest “Comparative analysis of shotgun and 16S rRNA metagenomics of gut microbiota in migratory seagulls”, lines 89-90, 98-99 that need to be reformulated, line 98 “creatures ” may be replaced by organisms.

All data, 8 bio-samples and sequences have been deposited in NCBI.

I suggest the manuscript should be better structured. For example, the authors can have a section for shotgun metagenomics, a second one for 16S rRNA metagenomics and a third one with statistical methods. As it is, the methods to estimate alpha-diversity (line 153), for example, are included in the section shotgun metagenomics but they were applied to taxa assignment by the two metagenomic methods. This kind of structure is confusing.

Based on the current manuscript quality, this manuscript could be accepted with major revisions.

Experimental design

The research is original since comparison between shotgun and 16S rRNA metagenomic techniques haven’t been done for seagulls gut microbiota. The research questions should be better defined. It is not clear what are the main objectives of the study: to compare the two metagenomic techniques or to compare two different places where wintering seagulls feed or, both. If the aim is to compare metagenomic methods, the authors should explain the importance of sampling two different places instead of using, for example, a mock community that could allow to estimate the accuracy of each method.

The figure legends need improvement. It would increase the quality of the paper if instead of a title the authors give a brief summary of what was done. The legends should be self-explanatory.

Unfortunately, the figures lost quality and can barely be seen.

Methods should be better detailed. I suggest the authors give more detail about the two sample sites, including the distance between the two; about the statistical methods that were used, including usage of rarefaction to adjust for the variations in metagenomic library sizes and the statistic tests used (in some cases more than one test is described in the methods (line 154, lines 156-157) but the results are not shown in the results section).

The authors used CheckM (line 163) to estimate contamination resulting from the use of shotgun metagenomics but they did not estimate possible contamination for the 16S rRNA metagenomics. It is not said if negative controls were used for DNA extraction and sequencing.

Analysis of results should be better explained. The authors should state what were the number of reads resulting from sequencing and how many reads were used for downstream analyses after quality control trimming, error and removal of chimeras. P-values or other statistical measure of association should be clearly stated in the text.

It is not clear why the authors have used the Welch’s-test (line 154) to compare alpha-diversity. These indices are known to be skewed distributions and the Welch’s-test assumes normality. On the other hand, the values the authors estimated are from the same sample but sequenced by two different methods. Paired sampled tests should be used. The authors also mentioned that they used PCA, PCoA and MDS (lines 156) but it is not clear where the results of these analysis are. A similar comment for other statistical tests like Adonis, Anosim and heatmaps (lines 157-158).
It is not clear what the authors meant by the top ten most abundant species in different taxonomic levels (lines 190-191). Shouldn’t the most abundant species be the same? Did the authors mean taxa?
It is not clear what the authors mean by “spatial distribution” when using Procrustes analysis (line 196). Please explain what data was used for this analysis.
Taxa names obtained directly from QUIIME should be changed, keeping, for example, the genus or other taxa to describe the taxonomy level of interest (Figure 1e, Figure 2c, Figure 3d).

In Figure 4 it seems that the P-value was higher than 0.05 and thus the authors fail to reject the null hypothesis. It is not clear which was the statistic test used (Welch’s t-test or Wilcoxon). Assuming the authors are comparing boxplots from 8 points, for each location, It is difficult to understand this result (P > 0.05) since in some cases boxplots (with equally n) do not overlap (which means the difference should be significant). Nonetheless, only 4 points are represented in the figure (which may be due to the low image quality). If indeed only 4 points are represented the test has zero-power.

Based on the experimental design quality this manuscript should be rejected.

Validity of the findings

The current discussion is focused on what the authors have found. However, the authors should link their observed results to the “bigger picture” of possible transmission of zoonoses on coastal areas where seagulls live in close proximity with humans.
In general, the discussion needs more detail. Specifically the authors could discuss why only a small proportion of common taxa, judging from figure 1 were identified by the two methods. What are the taxa that were identified by shotgun metagenomics but not by 16S rRNA (rare taxa?) and vice-versa?
For the 16S rRNA metagenome sequencing, the SILVA database was used to assign taxonomy. The SILVA database may not have represented the name of species that are rare and difficult to culture. Can some of the differences found by the authors be because the SILVA db is incomplete?

Given what the authors showed in the results I believe that they do not support the conclusions of the paper where the authors stated that the methods gave consistent results between the two (lines 308-310).

Based on the validity of the findings this manuscript should be rejected.

Additional comments

No comment.

Reviewer 3 ·

Basic reporting

Introduction
The introduction needs considerable work to provide a broader contact for this study. What about other studies that have compared 16s and metagenomic sequencing in different species? What general conclusions have been reached?

Line 103. Provide valid common name for Larus ridibundus.

Lines 103 to 108. Provide citations that this species is migratory and that densities are high in urban areas of China.

Experimental design

Methods
Sample Collection – How fresh were the samples? What were the samples stored in? How did you ensure that samples from the same individual were not collected multiple times?

Provide co-ordinates for the two sampling sites – HQ and HXQ.

Metagenomics
Lines 138-139. How many PCR cycles were performed for enrichment?

Line 143. Provide additional information on how the reads were filtered.

Line 153 to 154. What script was used given that Python is a language rather than a package?

16s
Line 177. How was the raw data trimmed and quality control implemented?
Lines 177 to 185. Provide citations for all these packages used.

What level of sequence similarity was used to classify an OTU, as all downstream analyses depend upon this?

Comparison
Provide citations for all the different packages used, at present several citations are missing.

Validity of the findings

Results
Lines 236 to 240 and Figure 3b. The text does not seem to agree with the figure; the 16s results do not seem to match the metagenomic results at all in comparison to the other hierarchical levels of taxonomic organization.

Lines 250 to 253. Please report the statistical results. I don’t understand how there are no significant differences given that in some of the plots in Figure 4 the differences between the two localities are so large that the 95% confidence intervals to not overlap. Please check this.

Additional comments

Discussion
Lines 287-188. I would say that 16s is used due to its low cost not high costs – i.e. as your next sentence implies.

Lines 302 and 303 – rather turn these values into percentages to be sure that readers interpret these results consistently.

One of the more useful applications of metagenomic approaches relative to 16s metabarcoding is the ability to identify bacterial down to strains. What did you discover with respect to the E. coli and Salmonella strains present? Both methods detected these bacteria, but the metagenomics should provide a lot more insight into considering whether these strains are a natural part of the gut flora or are these strains pathogenetic to either the bird, humans, or both. This aspect could be added to the discussion and would improve linkage of the discussion with the introduction.

This study really needs to be framed in a broader context. How do the results from this study fit into the broader context of work comparing 16s and metegenomic data. Right now the study is too descriptive to be of broad interest.

---

## Round 0.2 · Major Revisions

Dear Dr. Liao and colleagues:

Thanks for submitting your revised manuscript to PeerJ. I have now received only one review, but I cannot force the original reviewers to re-review.

Your paper still needs a bit of work. The reviewer provides many suggestions to help improve your work and manuscript (especially presentation).

Please deposit all your code on a public repository. Also, ensure your statistics are performed accurately.

Therefore, I am recommending that you revise your manuscript, accordingly, taking into account all the issues raised by the reviewer.

Good luck with your revision,

-joe

Reviewer 3 ·

Basic reporting

It’s not clear to me what the value is of citing the Rubiola et al. et al. paper. This does not seem to say anything beyond that metagenomic and 16s sequencing are correlated in terms of the diversity of bacterial species recovered. Why not find a paper (or system) that has used 16s and metagenomics, where the results of metagenomic studies have provided new insights into bacterial function beyond what the 16s data revealed.

Revise the text below as it is not a complete sentence.
“The significant correlation of the two methods for taxa diversity and richness, together with the similar profiles defined for both microbiomes (Rubiola et al. 2022).”

Line 150 – correct the spelling of communities.

Note the title 16s rRNA should be changed to 16s rDNA – i.e. ribosomal DNA. This should be changed throughout the manuscript.

16S rDNA sequencing – what does it mean that the samples were paired for library prep? You did two individuals pooled into a single lane? How many PCR were performed before library prep?

Provide citation for MetaGeneMark

Line 301 – what are unigenes?

Experimental design

Results - Annotations as bacteria in the metagenome were recalculated for the relative abundance of each taxon for comparison with 16S rRNA sequencing – this needs further elaboration. In the methods rarefraction is mentioned as having been used, but what was the threshold used?

Line 411 – change “species” to OTU. Note that in many places in the manuscript you are still using species when as the previous reviewer 1 pointed out you should really be using OTU or ASV.

Figure legends. I think some of the legends can be shortened and modified for clarity. For example, in Figure 1 - Comparison of the metagenome and 16S rDNA sequencing results for the number of bacterial taxa recovered by each method from fecal samples of the black-headed seagulls at the following levels of classification: phylum, class, order, family, OTU. (f) Cladogram of common and unique taxa composition between the two methods.

Figure 1f – this is the strangest cladogram I have seen. Surely it should reflect lineage diversity to some extent. I think this figure should be carefully checked.

Figure 2 could be moved to the supplementary documents.

Alpha Diversity results. I still do not understand why these results are not significantly different. Take 4e downwards for Simpson’s index. The 95% confidence intervals do not even overlap so how can these results not be significantly different? This same comment was mentioned previously by rev2 and rev3. This needs to be addressed in the manuscript, as it is very confusing. Why not use all 8 data points for these plots? If this cannot be addressed these plots should be remove as any reader is going to ask the same questions as the two previous reviewers.

Validity of the findings

With respect to the last paragraph before the conclusion section of your paper I suggest you review Choi et al. 2021 - High-throughput sequencing for examining Salmonella prevalence and pathogen-microbiota relationships in barn swallows https://doi.org/10.3389/fevo.2021.683183 --which suggests that 16s can be better than culture methods for detecting Salmonella, although I would agree with you that metabarcoding would be even better.

Additional comments

All used phyton scripts used in the python language should be submitted to dryad or github and a reference number provided.

---

## Round 0.3 · accepted · Accept

Dear Dr. Liao and colleagues:

Thanks for revising your manuscript based on the concerns raised by the reviewers. I now believe that your manuscript is suitable for publication. Congratulations! I look forward to seeing this work in print, and I anticipate it being an important resource for groups studying seagull microbiomes. Thanks again for choosing PeerJ to publish such important work.

Best,

-joe